# Sound Absorption Properties of DFs/EVA Composites

**DOI:** 10.3390/polym11050811

**Published:** 2019-05-06

**Authors:** Lihua Lyu, Yingjie Liu, Jihong Bi, Jing Guo

**Affiliations:** School of Textile and Material Engineering, Dalian Polytechnic University, Dalian 116034, China; 18840928558@163.com (Y.L.); bjhong888@163.com (J.B.)

**Keywords:** discarded feather fibers, hot-pressing, sound absorption properties, fractal dimension, composites

## Abstract

Using discarded feather fibers (DFs) and ethylene vinyl acetate (EVA) copolymer, the DFs/EVA composites with good sound absorption performance were prepared by hot-pressing method. The effects of hot-pressing temperature, mass fraction of DFs, density and thickness of composites on the sound absorption properties were studied by the controlling variable method. The sound absorption properties of the composites were studied by the transfer function method, and under the optimized technological conditions, the sound absorption coefficient of the composites was above 0.9 and the sound absorption band was wide. According to the box counting method based on the fractal theory, the fractal dimension of DFs/EVA sound absorption composites was calculated through Matlab programming, and the relationship between the fractal dimension and the mass fraction of DFs, the volume density of the composites were analyzed, then the quantitative relationship between the fractal dimension and the maximum sound absorption coefficient was deduced, which played a major role in the sound absorption design of porous sound absorption materials.

## 1. Introduction

Nowadays, one of the greatest concerns is the huge amounts of waste feathers that are produced year after year around the world [1]. These discarded feathers not only create a staggering waste, but also pollute the environment. In addition, those discarded feathers are more likely to cause flies and viruses that threaten human health. Thus, it is urgent to recycle these discarded feathers to make high value-added products.

Moreover, as the pace of industrialization and modernization grows larger, the noise pollution gets more serious. As acoustic buildings with important functions, such as recording studio, studio, cinema, etc., are also related to the design and installation of indoor acoustic materials [2]. In this sense, several research groups are working with discarded feather fibers (DFs) to obtain new materials with added value. From the point of acoustics, DFs are different from other natural fibers with large hollow structure, high porosity, low density, and good thermal and sound absorption properties [3,4,5]. In this regard, using DFs to develop sound absorption composites is an effective way to treat the DFs.

In recent years, many scholars have done a lot of research on the recycling of DFs. Carrillo et al. [6] established composites with DFs as reinforced material, high-density polyethylene, and polylactic acid as the matrix materials and the mechanical properties were investigated. When the content of DFs was 1–2%, the mechanical strength of the concrete was high, and when it exceeded 2%, the strength declined. Yang and Reddy [7] used high density polyethylene (HDPE) as matrix material and DFs as reinforced material and studied the sound absorption property of its composites. The study found that the composites made of DFs, as reinforced material, had better sound absorption performance than pure HDPE composites. Cynthia et al. [8] evaluated the performance as reinforcement of a fibrillar protein such as feather keratin fiber over a biopolymeric matrix composed of polysaccharides. These results demonstrated that chicken feathers could be useful to obtain novel keratin reinforcements and develop new green composites providing better properties, in comparison to the original biopolymer matrix. Dou et al. [9] developed and characterized mixed films based on FK (feather keratin) and PVA crosslinked by dialdehyde starch (DAS) for potential drug release applications. The obtained films had excellent compactness, compatibility, and water stability, which made it possible for the application of FK films in the field of biomaterials. Suhas et al. [10] used chicken feathers as filler to prepare composites with low load, high tensile, and bending strength, which could be applied to families, automobiles and structures. One author of this paper had been engaged in research on the sound absorption property of DFs and achieved some results. The author had tried to use discarded duck feathers and EVA by using lay-up and hot-pressing method to prepare a novel composite with good sound absorption properties. The noise reduction coefficient of novel composites could reach 0.76 [11]. However, the novel composites were not well formed during the study, and the sound absorption performance needed to be further improved.

This paper reported on the use of DFs as reinforcement in EVA powder composites with high sound absorption coefficient. Through single factor experiments, the effected of hot-pressing temperature, mass fraction of DFs, density of composites and thickness of composites on the sound absorption coefficient curve were analyzed emphatically. According to the self-similarity of DFs, the quantitative relationship between fractal dimension and maximum sound absorption coefficient was deduced by using the box counting method.

## 2. Experiment

### 2.1. Materials

EVA powder with 15 mesh and density of 0.95 g/cm^3^ (Plastic Raw Materials Management Department of Dongguan, Dongguan, China) and whole discarded feather fibers (DFs) with density of 1.14 g/cm^3^ (Poultry Farms of Dalian, Dalian, China) were used.

### 2.2. Equipment

WH-1 mixer (Shanghai Huxi Co. Ltd., Shanghai, China), QLB-50D/Q hot-pressing machine (Zhongkai Rubber Machinery Co. Ltd., Wuxi, China). Toledeo heat loss analyzer (METTLER Co. Ltd., Zurich, Switzerland), SW422/SW477 impedance tube sound absorption test system (Shengwang, Beijing, China) and JEOL JSM-6460LV SEM (JEOL Ltd., Beijing, China) were used.

### 2.3. Preparation of DFs/EVA Composites

The DFs/EVA sound absorption composites were prepared by hot-pressing method with whole DFs and EVA powder as raw materials. Whole DFs and EVA powder were mixed in a certain proportion using WH-1 mixer and heated with 8 MPa for 20 min using QLB-50D/Q hot-pressing machine [12]. Then, the DFs/EVA composites were formed and the test samples were disc-shaped composites with the size of Φ100 × 10 mm, Φ100 × 20 mm, Φ100 × 30 mm, and Φ100 × 40 mm.

### 2.4. Testing of the Composites

#### 2.4.1. Testing of Thermal Performance

Thermal performance test was carried out in the air flow rate of 50 mL/min, the heating rate of 10 °C/min and temperature range of 30–660 °C using the Toledeo heat loss analyzer. The weight loss curve of the whole DFs was obtained by one test.

#### 2.4.2. Testing of Sound Absorption Coefficient

Testing of sound absorption coefficient was done according to standard ISO10534-2. Under the conditions of atmospheric temperature 22 °C and relative humidity 68%, the sound absorption coefficient curves of the samples were tested by using SW422/SW477 impedance tube sound absorption test system. The measured sound absorption coefficient curve was the average of six measurements. The sound absorption coefficient curve test chart was shown in Figure 1.

#### 2.4.3. Calculation of Porosity

The porosity of DFs/EVA composites was tested using ASTM D2734 (test methods for void content of reinforced materials). The main formula was shown in Formula (1):(1)$$\varepsilon =100-\rho_{\alpha }\left(\frac{\gamma }{\rho_{e}}+\frac{g}{\rho_{f}}\right)$$

In the formula: $\rho_{\alpha }$ was the density of DFs/EVA composites (g/cm^3^), *r* was the mass fraction of EVA (weight %), $\rho_{e}$ was the density of EVA (g/cm^3^), g was the mass fraction of whole DFs (weight %), and $\rho_{f}$ was the density of whole DFs (g/cm^3^).

#### 2.4.4. Fractal Characterization

The surface morphology of seven composites samples were scanned by JEOL JSM-6460LV SEM, and the pixel size of the intercepted images were 1024 × 1024. The image processing software Photoshop was used to process the gray level of the obtained scanning electron microscope images and the image threshold was set. Then, the image with gray level was converted into a binary image that could be recognized by the computer, and the box counting method [13] was used to calculate the fractal dimension of the sample by Matlab programming (the procedure was shown in Appendix A).

The calculation principle of the box counting method was to take a small cube box with an edge length of ε and cover the curve graph with fractal characteristics. Some boxes were empty, some boxes had a part of the curve, and the number of boxes containing the curve was recorded as N (ε). Then, shorten the size of the box, and N (ε) would increase correspondingly. When ε was close to 0, the fractal dimension of the curve was obtained:(2)$$D=-\underset{\varepsilon \to 0}{\lim}\frac{\log N\left(\varepsilon \right)}{\log\varepsilon }$$

However, in the actual calculation process, the value of *ε* was not infinite, but a finite value. By fitting and mapping with the least square method, a straight line was obtained, and the fractal dimension was the slope of the straight line.

## 3. Results and Discussion

### 3.1. Effect of Hot-Pressing Temperature on Sound Absorption Coefficient

The properties of the whole DFs changed under the thermal action. At the optimum hot-pressing temperature, the performance of EVA was best and the performance of whole DFs would not be changed. The sound absorption performance of the DFs/EVA composites would be improved accordingly. In order to determine the optimal hot-pressing temperature, the weight loss curve of the whole DFs was studied during the temperature change and the weight loss curve of the whole DFs was shown in Figure 2.

As shown in Figure 2, thermal decomposition temperature of the whole DFs was 226.9 °C. When the temperature was in the range of 46.5 °C to 102 °C, the physical adsorption water of the whole DFs was vaporized. Then, when the temperature was in the range of 102 °C to 226.9 °C, the chemical combined water was removed. In this process, the surface color of the whole DFs began to yellow from about 130 °C, its physical properties began to produce irreversible changes. So, the hot-pressing temperature of the DFs/EVA composites should be controlled below 130 °C. The melting point of EVA was 77 °C [14]. Therefore, the range of the hot-pressing temperature was 90–120 °C.

In the appropriate technological conditions and under the 90, 100, 110, and 120 °C hot-pressing temperature, respectively, DFs composites were prepared. The effect of hot-pressing temperature on the sound absorption properties of the DFs composites was shown in Figure 3. From Figure 3, with the increase of the frequency, the four kinds of test samples at the beginning (at 0 to 500 Hz) had a little influence on the sound absorption coefficient and the sound absorption properties of the beginning were not good, the maximum absorption coefficient was only 0.16. Then the sound absorption coefficients of test samples showed the growth trend in the range of 500–4000 Hz. However, the sound absorption coefficients dropped slightly in the range of 4000–6400 Hz. Within the entire test frequency range, the absorption coefficients of the samples decreased with the increase of temperature. When the hot-pressing temperature was 90 °C, the sound absorption properties of sample were the best, the maximum sound absorption coefficient reached 0.54.

With the increase of temperature, the dehydration and carbonization of DFs affected its own acoustic performance, which leaded to the decrease of absorption properties. When the hot-pressing temperature was too high, the thermal degradation and collapse of the samples could be caused, resulting in a sharp drop on the absorption properties. When the hot-pressing temperature was 90 °C, EVA had reached the best softening melting point, its liquidity and diffusivity were the best. A fine mesh pore structure could be formed between DFs and EVA. Therefore, the optimal hot-pressing temperature was 90 °C.

### 3.2. Effect of Mass Fraction of DFs on the Sound Absorption Coefficient

In order to study the effect of the mass fraction of DFs on sound absorption properties, samples of DFs composites with mass fractions of 20%, 30%, 40%, 50% and 60% were obtained, and its sound absorption coefficients were determined under the optimum conditions. Figure 4 showed the influence of the mass fraction of DFs on sound absorption coefficient. Table 1 was maximum sound absorption coefficient and average sound absorption coefficient of the DFs composites with different mass fraction, where the average sound absorption coefficient was the arithmetic mean of the absorption coefficients in 125 Hz, 250 Hz, 500 Hz, 1000 Hz, 2000 Hz, and 4000 Hz [15].

Previous reports revealed that the amount of fibers in the composites had significant effect on the properties of the composites and the properties were maximum at the optimized fiber amount [16]. It could be seen from Figure 4 that as the mass fraction of DFs had been increased, the acoustic performance had increased. In addition, the sound absorption properties of each sample were poor at 0–500 Hz, but the sound absorption effect was excellent at 500–4000 Hz and decreased gradually after 4000 Hz. Among them, the sound absorption coefficient reached a peak at 4000 Hz. Table 1 showed that the sound absorption coefficient of the samples increased within a certain range with the increase of mass fraction of DFs. When the mass fraction of DFs was 60%, the maximum sound absorption coefficient of sample was 0.69. When the right amount of DFs was added, sound energy could be dissipated through the vibration of the fiber to improve the sound absorption properties of the composites. The experimental results showed that the composites had high loftiness, low breaking strength, and the forming effect of composites was not good when excess DFs were added.

Due to the special morphological characteristics of feathers [17], the barbs and the joints of the bone overlapped each other through the adhesion of EVA, thus the sample had more holes. In a certain range, when the mass fraction of DFs increased, the number of fibers in the unit volume increased, resulting in a large number of micropores and good sound absorption performance. Besides, a small number of fibers mean a high content of EVA hot melt adhesive resin and too many bonding points. The bonding points made the DFs tight to each other and caused DFs lost the support function. Therefore the absorption effect was poor with a little number of pores. Based on the above analysis, mass fraction of DFs was 60%.

### 3.3. Effect of Density of DFs Composites on Sound Absorption Coefficient

Under these optimized technological conditions, samples of 4 different densities of 0.2 g/cm^3^, 0.3 g/cm^3^, 0.4 g/cm^3^ and 0.5 g/cm^3^ were prepared. The porosity of the DFs composites were calculated as shown in Figure 5. As could be seen from Figure 5, when the density of the sample increased from 0.2 g/cm^3^ to 0.5 g/cm^3^, the porosity of the sample drops from 81.1% to 52.6% with other conditions remain unchanged. It showed that the change of density had remarkable influence on the porosity of composites. The reason might be that when the density increased, the materials in the unit volume were more compact and the cavity was less, therefore the porosity of the composite materials was reduced. To further investigate the effect of density on sound absorption properties, the sound absorption coefficient of composites under different densities were tested. The sound absorption coefficient curves obtained were shown in Figure 6.

From the results of Figure 6, the sound absorption performance of the composites first increased and then decreased with the decrease of the composites density. It could be seen that there was an extreme value of its density to obtain the best sound absorption performance. When the density of the composites was 0.3 g/cm^3^, the maximum sound absorption coefficient was 0.91, and the absorption band was the widest. This was because that the density of the composites was small, the material was given an appropriate number and size of pores, and most of the sound waves were absorbed by the composites. It was not the smaller the density of the composites, but the best sound absorption property when the density of the composites reached the most appropriate value. As the volume density was too small, the holes became very large, and the sound wave could directly penetrate through the material, thus the sound absorption effect could not be achieved. When the density of the composites further increased, the number of pores was dropping. The sound waves were not easy to enter the interior of the composites and most of the sound waves were directly reflected by it. So the sound absorption performance was deteriorated [18]. This low absorption coefficient for high densities could also be explained by the low porosity and the diminution of the size of the pores, which were key parameters in sound absorption [19]. Therefore, too low or too high density would affect the sound absorption properties of the composites. There was a relatively suitable density range to make the sound absorption effect of composites the best. Based on the above analysis, the density of the composites was 0.3 g/cm^3^.

### 3.4. Effect of Thickness of DFs Composites on Sound Absorption Coefficient

Material thickness is one of the factors that reflect its sound absorption performance, and it is one of the ways to improve the sound absorption performance of the materials by increasing the material thickness [20]. Figure 7 showed the effect of thickness of DFs composites on sound absorption coefficient. Table 2 showed the sound absorption parameters of specimens of different thicknesses.

It could be seen from Figure 7 that, with the increase of the thickness of the composites, the sound absorption properties of the middle and low frequency band were obviously improved, the sound absorption coefficient of the high frequency band did not change or even decreased after reaching a certain value, and the peak absorption coefficient moved an octave in the direction of low frequency [21]. When the thickness was 20 mm, the peak frequency of sound absorption coefficient was 1600 Hz, and when the thickness increased to 30 mm, the peak frequency moved forward an octave. The average absorption coefficient of composites improved as the thickness increased. Due to the spread of sound waves inside the porous material, paths lengthened with the increase of the thickness of the materials. At this point, the acoustic wave induced viscosity loss and thermal conduction was more abundant, which was especially beneficial to low-frequency acoustic waves with longer wavelength. However, with the increase of the thickness, acoustic impedance difference between material medium and air medium got bigger. It enhanced the reflection of the materials by the sound wave. Thus, the sound wave entering the materials was reduced. Therefore, the sound absorption property of high frequency acoustic wave had declined. According to the average sound absorption coefficient of the composites in Table 2, when the thickness of the material was 4 mm, the material had excellent low-frequency sound absorption performance and the highest average sound absorption coefficient. Based on the above analysis, the thickness of the composites was 4 mm.

### 3.5. Fractal Characterization Results

Discarded feathers met the core characteristics of fractal theory in morphological characteristics and had strictly statistical self-similarity [22]. Self-similarity was a typical property of fractal objects: at any scale one could find a small piece of the object that was similar to the whole in a deterministic or stochastic sense [23]. Therefore, the sound absorption performance of discarded feathers could be solved by the fractal method. Figure 8 was a self-similarity characteristic diagram of discarded feathers.

Among the relevant influencing factors of DFs/EVA sound absorption composites, the mass fraction of DFs and density of composites could best reflect its own structure. Therefore, the relationship between fractal dimension and the above two factors was established. The SEM images of the surface morphology of samples numbered sample 1 to sample 7 were shown in Figure 9.

Because the contrast of the images were too low, it was difficult to distinguish the fibers from the pores. The software Photoshop was used to process the gray level of the scanning electron microscope of the samples, so that the brighter the bright place was, the darker the dark place was in the whole image range, thus achieving the purpose of enhancing the images. Then, set the images threshold and convert the images with gray scale into a binary image, finally binary images recognized by the computer were obtained. The pre-processed images were shown in Figure 10.

According to the Matlab program programmed by the box counting method, the obtained pre-processed images were calculated to obtain the fractal dimension of the samples. The calculation results of the sample fractal dimension were shown in Figure 11.

The slope of each straight line was the fractal dimension of the corresponding sample. After obtaining the fractal dimensions of the DFs/EVA sound absorption composites samples, the relationship between the fractal dimension and the basic structural parameters was analyzed. Relevant parameters of the samples were shown in Table 3.

Table 3 showed the basic parameters of the 7 samples. It could be seen that the correlation degree of the 7 samples was above 0.9, indicating that the fractal characteristics of the samples were very obvious.

#### 3.5.1. Relationship between Fractal Dimension and DFs Mass Fraction

Figure 12 showed the relationship between fractal dimension and the mass fraction of DFs. It could be seen from the figure that with the increase of DFs mass fraction, its corresponding fractal dimension also increased, and was approximately linearly correlated. The variation trend between the mass fraction of DFs and the fractal dimension of the sample and the sound absorption performance of the material was approximately similar.

#### 3.5.2. Relationship between Fractal Dimension and Density of DFs Composites

Figure 13 showed the relationship between fractal dimension and density of DFs composites. As it could be seen from the figure, with the increase of sample density, the corresponding fractal dimension increased first and then decreased. The fractal dimension reached the maximum value under the optimal sample density.

#### 3.5.3. Relationship between Fractal Dimension and Sound Absorption Coefficient

As shown in Figure 14, the relationship between the fractal dimension of 7 samples and the maximum sound absorption coefficient was shown. It could be seen from the figure that the fractal dimension of the material had a certain relationship with the maximum sound absorption coefficient. That was, the larger the fractal dimension was, the bigger the maximum sound absorption coefficient was.

In order to obtain the quantitative relationship between the fractal dimension and the maximum sound absorption coefficient, the fractal dimension and the maximum sound absorption coefficient were fitted, and the fitting curve was shown in Figure 15.

The fitting relation was as follows:(3)$$Y=17.037X^{2}-58.261X+50.153$$

In the formula: Y was the maximum sound absorption coefficient; X was the fractal dimension.

The correlation coefficient between the fractal dimension and the maximum sound absorption coefficient was 0.952, which showed that there was a strong positive correlation between the fractal dimension and the maximum sound absorption coefficient. In the design of porous sound absorbing materials, the maximum sound absorption coefficient of the sound absorbing materials could be predicted through the fitting relation obtained above.

## 4. Conclusions

DFs/EVA composites with high porosity were prepared by hot-pressing method with DFs and EVA. Additionally, the following conclusions were drawn:

(1) Through single factor experiments, the results showed that the optimized technological conditions were as follows: hot-pressing temperature 90 °C, mass fraction of DFs 60%, density of composites 0.3 g/cm^3^, and thickness of composites 4 mm. Therefore, the porous sound absorbing materials with high sound absorption coefficient and wide absorption band was obtained and the maximum sound absorption coefficient was above 0.9.

(2) The fractal dimension of DFs/EVA sound absorption composites was calculated by using box counting method. The results showed that the DFs/EVA sound absorption composites had strong fractal characteristics, and the fractal dimension and the sound absorption performance of the material were similar with the change trend of DFs mass fraction. With the increase of volume density, the corresponding fractal dimension increased first and then decreased, and the fractal dimension reached the maximum value under the optimal volume density. The quantitative relationship between the fractal dimension and the maximum sound absorption coefficient obtained by fitting was Y = 17.037X^2^ − 58.261X + 50.153, and the correlation coefficient was 0.952, which played a theoretical basis and guiding role in the sound absorption performance design of porous sound absorption materials.

## Figures and Tables

**Figure 1 polymers-11-00811-f001:**
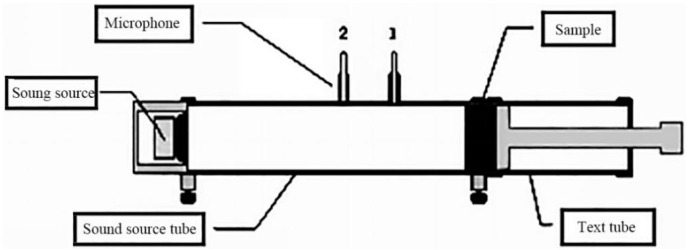
Schematic diagram of sound absorption test.

**Figure 2 polymers-11-00811-f002:**
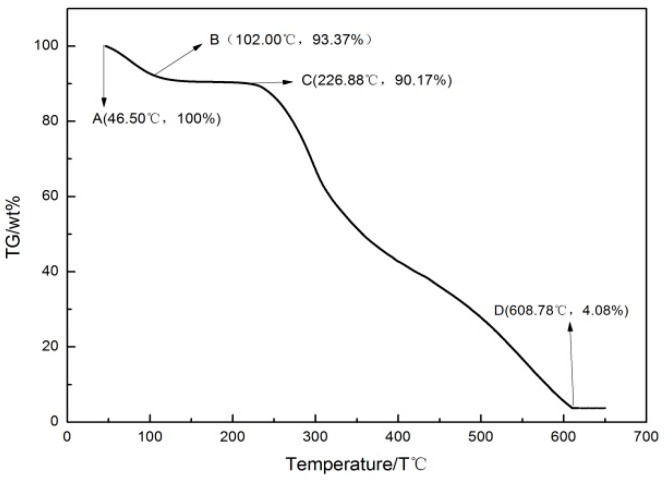
The weight loss curve of the whole DFs.

**Figure 3 polymers-11-00811-f003:**
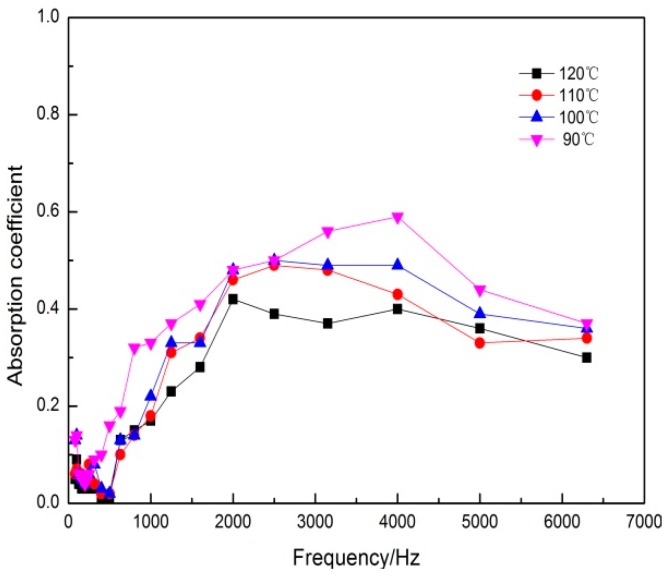
Effect of hot-pressing temperature on sound absorption coefficient.

**Figure 4 polymers-11-00811-f004:**
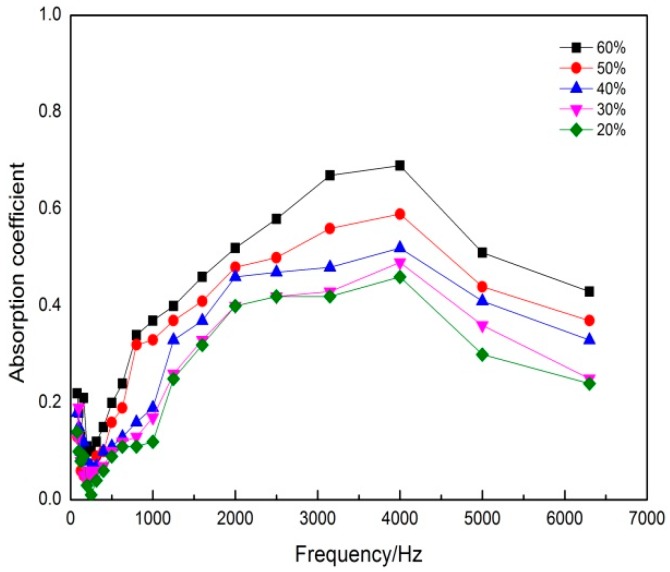
Effect of the mass fraction of DFs on sound absorption coefficient.

**Figure 5 polymers-11-00811-f005:**
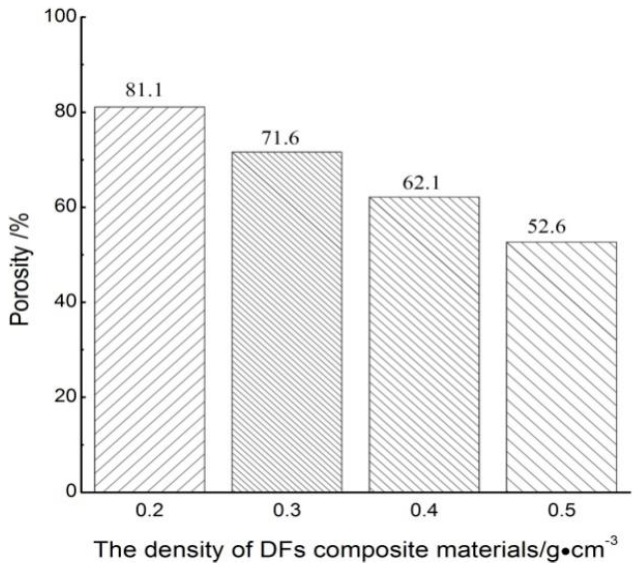
Porosity of the composites with different densities.

**Figure 6 polymers-11-00811-f006:**
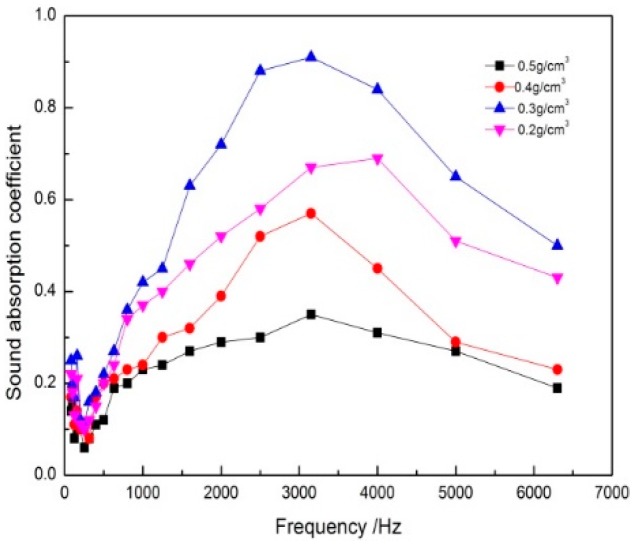
Effect of density of composites on sound absorption coefficient.

**Figure 7 polymers-11-00811-f007:**
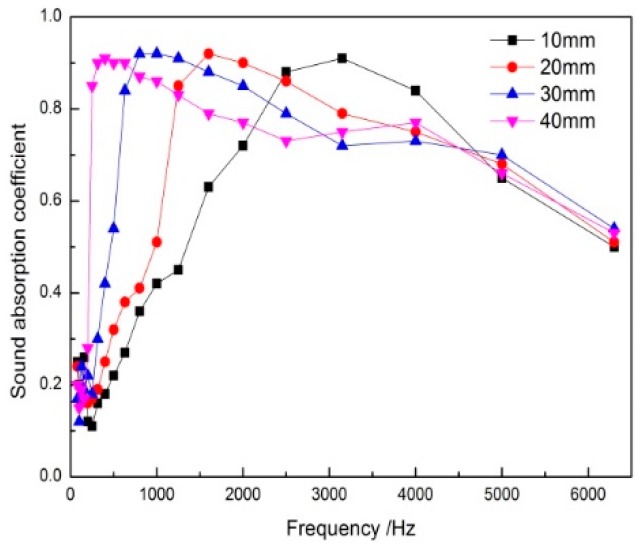
Effect of thickness of composites on sound absorption coefficient.

**Figure 8 polymers-11-00811-f008:**
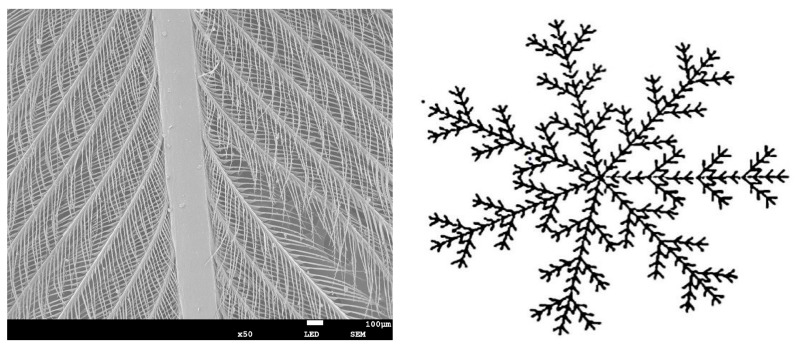
Self-similarity characteristic diagram of discarded feathers.

**Figure 9 polymers-11-00811-f009:**
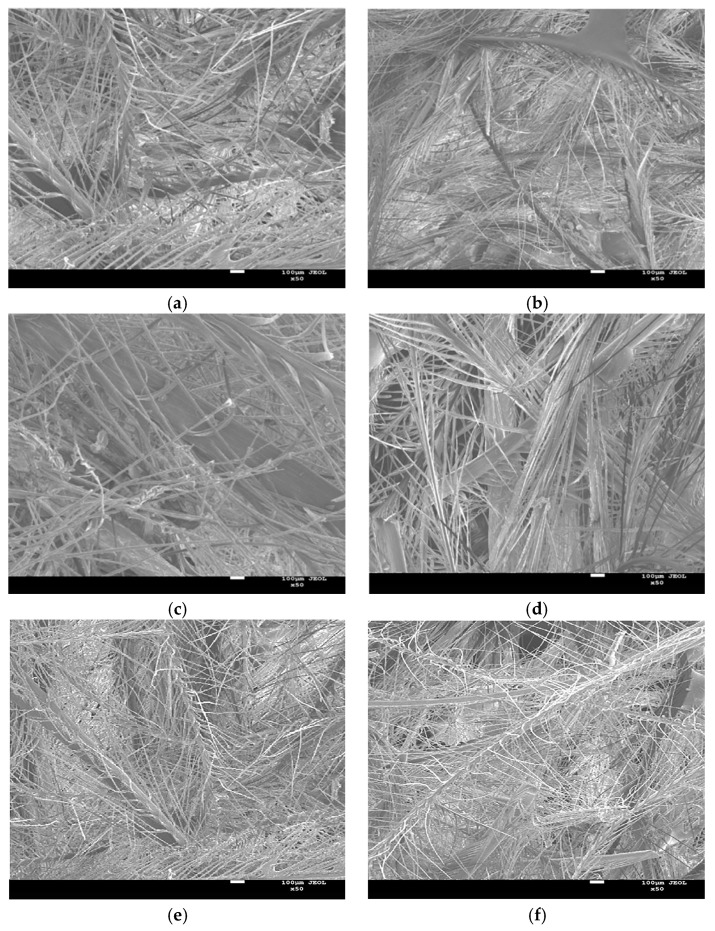
SEM images of samples (#1–7) surface morphology (**a**–**g**).

**Figure 10 polymers-11-00811-f010:**
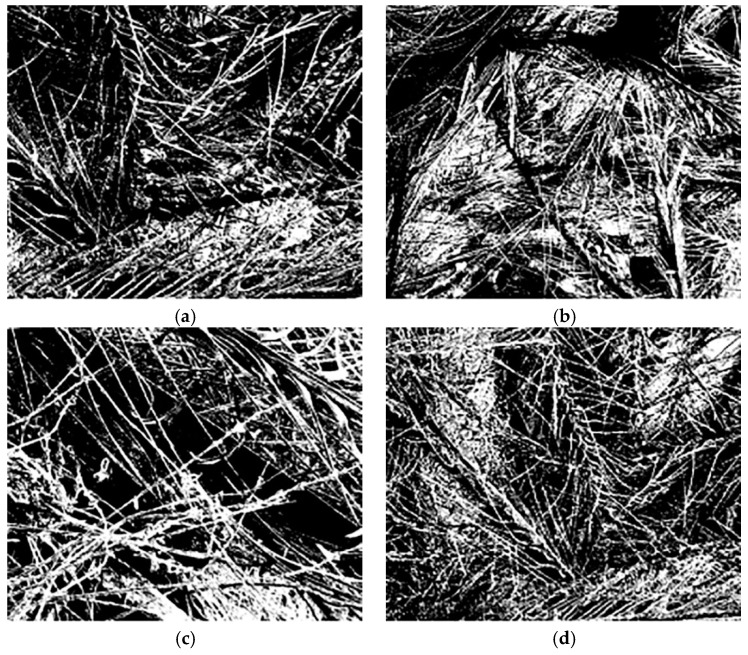
Images of pre-processed samples (#1–7) (**a**–**g**).

**Figure 11 polymers-11-00811-f011:**
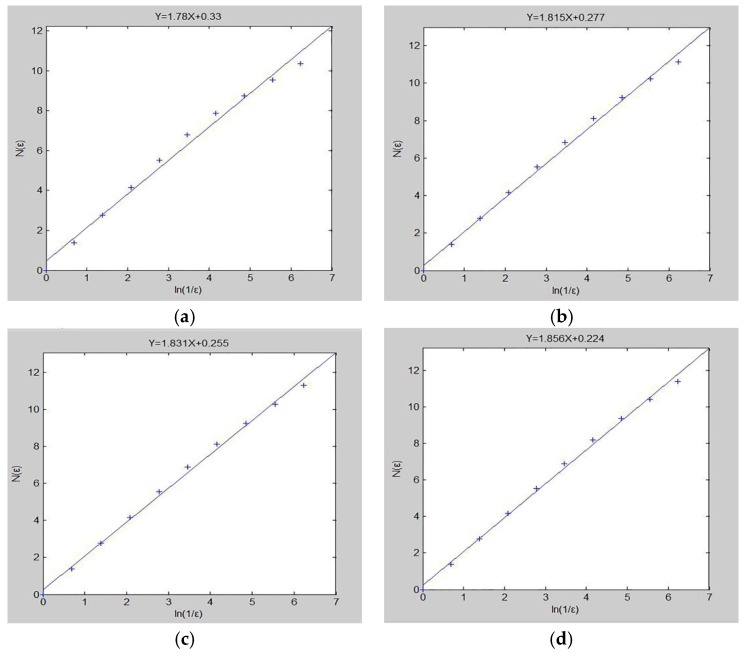
Calculation results of fractal dimension of samples (#1–7) (**a**–**g**).

**Figure 12 polymers-11-00811-f012:**
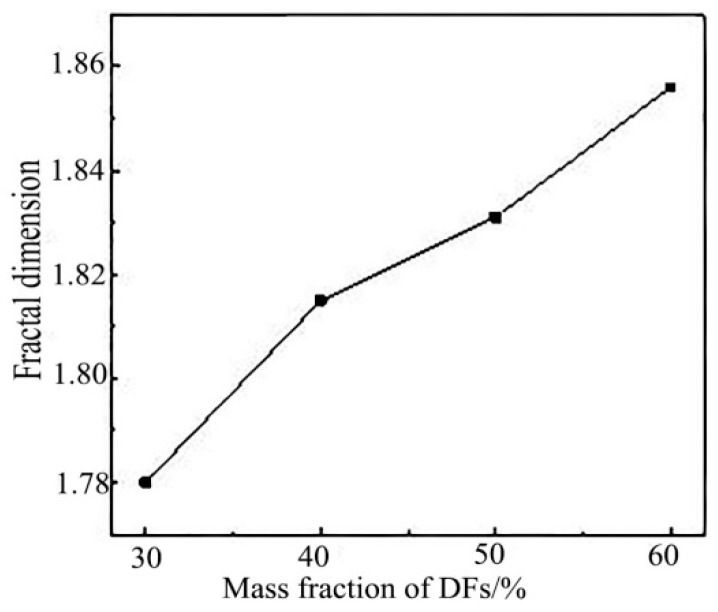
Relationship between fractal dimension and mass fraction of DFs.

**Figure 13 polymers-11-00811-f013:**
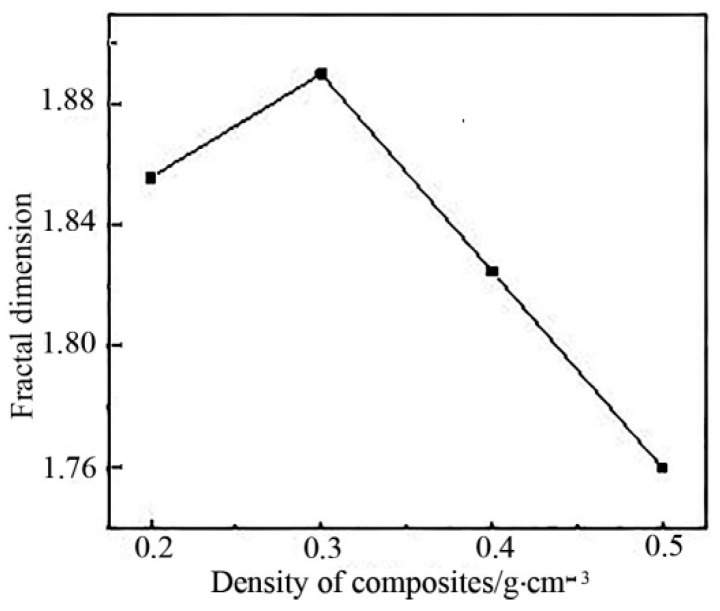
Relationship between fractal dimension and density of DFs composites.

**Figure 14 polymers-11-00811-f014:**
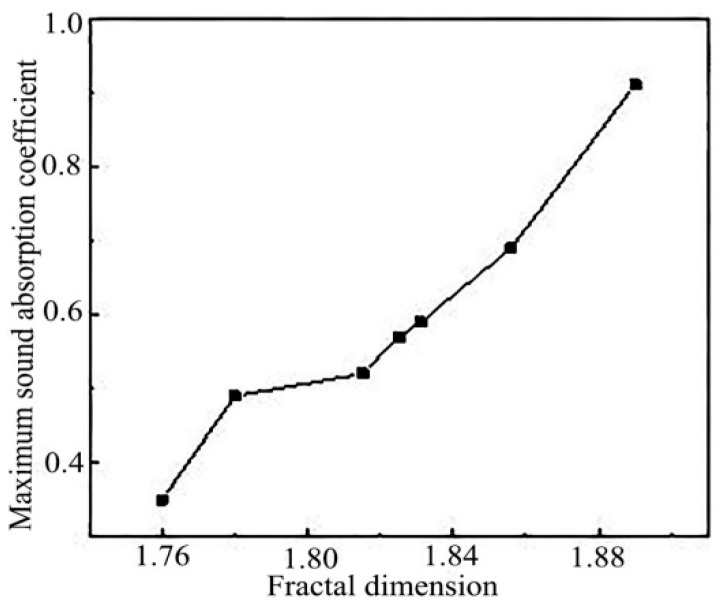
Relationship between fractal dimension and maximum sound absorption coefficient.

**Figure 15 polymers-11-00811-f015:**
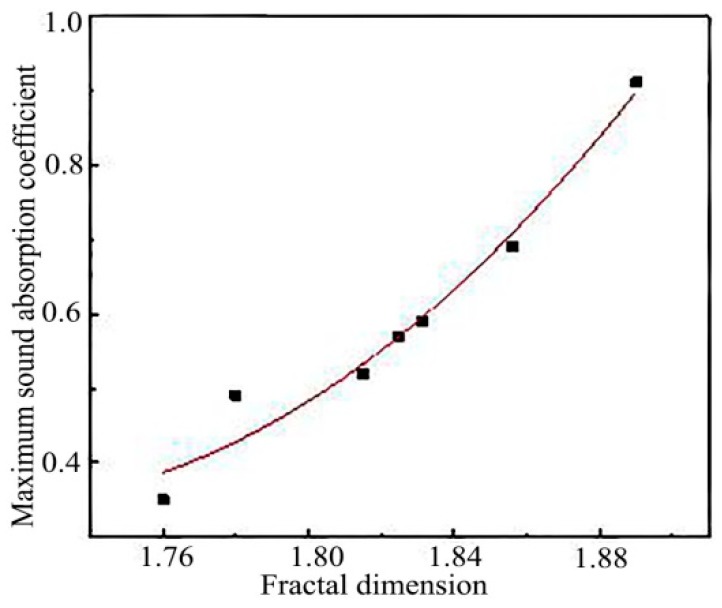
Fitting curve between fractal dimension and maximum sound absorption coefficient.

**Table 1 polymers-11-00811-t001:** Maximum sound absorption coefficient and average sound absorption coefficient of the DFs composites with different mass fraction.

Mass Fraction of DFs (%)	Maximum Sound Absorption Coefficient	Average Sound Absorption Coefficient
20	0.46	0.193
30	0.49	0.215
40	0.52	0.242
50	0.59	0.280
60	0.69	0.335

**Table 2 polymers-11-00811-t002:** Sound absorption parameters of specimens of different thicknesses.

Sample	Thicknesses/mm	Frequency of Sound Absorption Coefficient Peak/Hz	Average Sound Absorption Coefficient
1	10	3150	0.41
2	20	1600	0.49
3	30	800	0.58
4	40	400	0.72

**Table 3 polymers-11-00811-t003:** Relevant parameters of samples.

Sample Number	Mass Fraction of DFs/%	Density of DFs/g·cm^−3^	Maximum Sound Absorption Coefficient	Fractal Dimension	Correlation Coefficient
1	30	0.2	0.49	1.780	0.99288
2	40	0.2	0.52	1.815	0.99519
3	50	0.2	0.59	1.831	0.99616
4	60	0.2	0.69	1.856	0.99658
5	60	0.3	0.91	1.890	0.99757
6	60	0.4	0.57	1.825	0.99457
7	60	0.5	0.35	1.760	0.99219

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
