# Peer review of "Sound Absorption Properties of DFs/EVA Composites"

_polymers, 2019, doi:10.3390/polym11050811_

Round 1

Reviewer 1 Report

The submitted work reported a hot-pressing method to prepare DFs/EVA composites with excellent sound absorption properties. The effect of hot-pressing temperature, amount of DFs, density of composites and thickness of DFs/EVA composites were comprehensively investigated. The data presented are of high quality and the manuscript can be accepted for publication after adding some sort of discussion to compare the current system with the state of the art. Some references are suggested for cited, e.g. Macromolecular Materials and Engineering, 2018, 1800336; Macromolecular Materials and Engineering, 2017, 302(1), 1600353; Polymer Chemistry 2018, 9(20), 2685-2720, etc.

Author Response

Response: I am very grateful to the reviewer for the constructive comments. New reference have been cited. Previous reports revealed that the amount of fibers in the composites had significant effect on the properties of the composites and the properties were maximum at the optimized fiber amount[16].There is in page 5-168.

Reviewer 2 Report

The paper presents an interesting work on the sound absorption characterization of polymer composites reinforced by discarded feather fibers. The topic is interesting and the paper deserves publication after minor revisions, listed below.

Page 3: the number of replicates for each kind of experiemtn should be indicated.

DFs/EVA Composites: please indicate more clearly the sample geometry and dimensions.

Testing of Sound Absorption Coefficient: more details on the test specimens, the measurement method should be added. Are the samples in a form of disc with a diameter of 100 mm, as I guess from row 92? Since this test is the most important for the discussion of the results, the measurement method should be explained also by the addition of a scheme or a picture.

Page 3: use only one digit for the temperatures.

Figure 4: use only one digit

Page 10: it is not clear the procedure for obtaining the fractal dimensions starting from the pre-processed images of page 10. Explain the y-axis and x-axis of the plots in Figure 10 and how they have been obtained.

The word weightlessness should be changed by weight loss overall the text.

Author Response

Page 3: the number of replicates for each kind of experiment should be indicated.

Response: According to the suggestion, the number of replicates for each kind of experiment has been added. There is in page 3-90, 96.

DFs/EVA Composites: please indicate more clearly the sample geometry and dimensions.

Response: I'm sorry for the errors in the preparation of the article. The geometry and dimensions of the samples has been added. The test samples were a disc-shaped composites with the size ofФ100* 10mm,Ф100* 20mm,Ф100* 30mm and Ф100* 40mm. There is in page 2-85. 

Testing of Sound Absorption Coefficient: more details on the test specimens, the measurement method should be added. Are the samples in a form of disc with a diameter of 100 mm, as I guess from row 92? Since this test is the most important for the discussion of the results, the measurement method should be explained also by the addition of a scheme or a picture.

Response: According to the suggestion, the geometry and dimensions of the samples has been added. The test samples were disc-shaped composites with the size of Ф100* 10mm,Ф100* 20mm,Ф100* 30mm and Ф100* 40mm.There is in page 2-85. 

The measurement method has been explained by a picture. There is in page 3-98.

Page 3: use only one digit for the temperatures.

Response: We have made correction according to the reviewer’s comments. There is in page 4-132, 134.

Figure 4: use only one digit

Response: According to the suggestion, the number in figure 4 took one digit. There is in page 6-204.

Page 10: it is not clear the procedure for obtaining the fractal dimensions starting from the pre-processed images of page 10. Explain the y-axis and x-axis of the plots in Figure 10 and how they have been obtained.

Response: We have made correction according to the reviewer’s comments. The box calculation principle has been added. There is in page 3-114.

The word weightlessness should be changed by weight loss overall the text.

Response: According to the suggestion, “weightlessness” in the paper has been replaced by “weight loss”. There is in page 4-127,128,131.

Reviewer 3 Report

The manuscript titled “Sound Absorption Properties of DFs/EVA  Composites” by Lyu Lihua, Liu Yingjie, Bi Jihong and Guo Jing, proposes the use of discarded feather fibers (DFs) and ethylene vinyl acetate (EVA) copolymer as sound absorber.They have conducted studies to analyze the influence of hot-pressing temperature, mass fraction of DFs, density of composites, the fractal dimensions and the thickness of composites on the sound absorption properties.

General remarks

This paper contains new data relating the fractal dimension to the sound absorption coefficient. Therefore it  merits publication but  before being published,  the poor English language (this does not deny this paper the merit of good scientific ideas) needs polishing as the paper is well written and the objectives clearly defined.  The only missing point is the determination of the intrinsic micro-geometric parameters, responsible for sound absorption, from the absorption coefficient data (see Mohamed Ben Mansour, Erick Ogam, Ahmed Jelidi, Amel Soukaina Cherif, Sadok Ben Jabrallah, Influence of compaction pressure on the mechanical and acoustic properties of compacted earth blocks: An inverse multi-parameter acoustic problem, Applied Acoustics, Volume 125, 2017, Pages 128-135, ISSN 0003682X, https://doi.org/10.1016/j.apacoust.2017.04.017). 

Using the maximum value of the absorption coefficient is a good idea as that was the final objective to be attained. However,  optimization  using different intrinsic acoustic parameters and the fractal dimensions recovered from the absorption spectrum  can lead to a better and faster way of designing the sound absorption packages i.e. tailoring the material to have the  most pertinent parameter affecting the absorption coefficient. A Box counting program for determining the fractal dimension of samples from their binary images  has been provided in the appendix and congratulate the authors for this.It is a pity that a theoretical model for the absorption coefficient was not developed in this study. Models of acoustic wave propagation in fractal media can be found in (1) Vasily E. Tarasov, Acoustic waves in fractal media: Non-integer dimensional spaces approach, Wave Motion, Volume 63, 2016, Pages 18-22, ISSN 0165-2125, https://doi.org/10.1016/j.wavemoti.2016.01.003,  (2) A. Berbiche, M. Fellah, Z.E.A. Fellah, E. Ogam, F.G. Mitri, C. Depollier, Transient acoustic wave in self-similar porous material having rigid frame: Low frequency domain, Wave Motion, Volume 68, 2017, Pages 12-21, ISSN 0165-2125, https://doi.org/10.1016/j.wavemoti.2016.07.015). Please cite appropriately.

Detailed remarks

1.      Please unify the addresses of the authors. They are all apparently  from the same university.

2.      Page 1, line 29, please use another phrase instead of “reference role” -> major role?

3.      Page 1,  Line 37, please remove the word “seriously”.

4.      Page 1,  Line 41 please replace the phrase “noise pollution gets more and more serious” .

5.      Page 1,  Line 45 please remove the second  “good”  .

6.      Page 2, line 64  please modify to “One of the authors of this paper …” .

7.      Page 2,  Line 80 please add country to Dongguan.Same remark at Line 86 after Beijing.

8.      Page 5, line 164, please replace improved with increased.

9.      Page 6, line 208 please remove “and consumed”.

10.  In all the text need to be combed for typos and the English language polished by a native English speaker.

Author Response

Response: I am very grateful to the reviewer for the constructive comments. The article you provided has high reference value and has already been quoted in this article. I will learn the model and I hope it can be applied to future experiments. I revised them and There was in page 7-221, 8-255.

Detailed remarks

1. Please unify the addresses of the authors. They are all apparently from the same university.

Response: According to the suggestion, the address of the authors has been unified. There is in page 1-5.

2. Page 1, line 29, please use another phrase instead of “reference role” -> major role?

Response: According to the suggestion, “reference role” in the paper has been replaced by “major role”. There is in page 1-23.

3.   Page 1, Line 37, please remove the word “seriously”.

Response: I'm sorry for this error in the preparation of the article. The word “seriously” has been removed. There is in page 1-31.

4.   Page 1, Line 41 please replace the phrase “noise pollution gets more and more serious”.

Response: According to the suggestion, the phrase “noise pollution gets more and more serious” has been replaced by the noise pollution gets more serious”. There is in page 1-34.

5.   Page 1, Line 45 please remove the second “good”.

Response: According to the suggestion, the word “good has been removed. There is in page 1-39.

6.   Page 2, line 64 please modify to “One of the authors of this paper …”.

Response: As suggested by the reviewer, “The authors of this paper …” has been replaced byOne of the authors of this paper …”.There is in page 2-58.

7.   Page 2, Line 80 please add country to Dongguan. Same remark at Line 86 after Beijing

Response: According to the suggestion, the country has been added in the Materials part .There is in page 2-74, 75.

8.   Page 5, line 164, please replace improved with increased.

Response: As suggested by the reviewer, “improvedhas been replaced byincreased.There is in page 5-170.

9.   Page 6, line 208 please remove “and consumed”.

Response: I'm sorry for this error in the preparation of the article. The word “and consumed” has been removed. There is in page 7-214.

10.  In all the text need to be combed for typos and the English language polished by a native English speaker.

Response: I'm sorry for the errors in the preparation of the article. This paper has been checked carefully and some problems have been revised. There is in page 1 to16.

Round 2

Reviewer 3 Report

The authors have revised the article appropriately.